# Temperature/Reduction Dual Response Nanogel Is Formed by In Situ Stereocomplexation of Poly (Lactic Acid)

**DOI:** 10.3390/polym13203492

**Published:** 2021-10-12

**Authors:** Wenli Gao, Zhidan Wang, Fei Song, Yu Fu, Qingrong Wu, Shouxin Liu

**Affiliations:** Key Laboratory of Applied Surface and Colloid Chemistry, Ministry of Education, School of Chemistry and Chemical Engineering, Shaanxi Normal University, Xi’an 710119, China; 17853465526@163.com (W.G.); 18189608643@163.com (Z.W.); 17563713257@163.com (F.S.); fy2247127430@163.com (Y.F.); wuqingrong123456@snnu.edu.cn (Q.W.)

**Keywords:** stereocomplexation, polylactic acid, temperature/reduction, self-recombination

## Abstract

A novel type of dual responsive nanogels was synthesized by physical crosslinking of polylactic acid stereocomplexation: temperature and reduction dual stimulation responsive gels were formed in situ by mixing equal amounts of PLA (Poly (Lactic Acid)) enantiomeric graft copolymer micellar solution; the properties of double stimulation response make it more targeted in the field of drug release. The structural composition of the gels was studied by proton nuclear magnetic resonance (^1^H NMR) and Fourier transform infrared spectroscopy (FT-IR). Using transmission electron microscope (TEM) and dynamic light scattering (DLS) instruments, the differences in morphology and particle size were analyzed (indicating that nanogels have dual stimulus responses of temperature sensitivity and reduction). The Wide-Angle X-ray diffractionr (WAXD) was used to prove the stereocomplexation of PLA in the gels, the mechanical properties and gelation process of the gels were studied by rheology test. The physically cross-linked gel network generated by the self-recombination of micelles and then stereo-complexation has a more stable structure. The results show that the micelle properties, swelling properties and rheological properties of nanogels can be changed by adjusting the degree of polymerization of polylactic acid. In addition, it provides a safe and practical new method for preparing stable temperature/reduction response physical cross-linked gel.

## 1. Introduction

In the field of drug delivery, there are many types of polymer-based carriers, such as nanoparticles [1], vesicles [2,3], polymer micelles [4,5,6,7], hydrogels [8,9,10], etc. Among these carriers, polymeric micelles are one of the most common. However, the instability of the micelle structure causes it to dissociate at a concentration lower than the critical micelle concentration, which will lead to the loss of drugs during the blood circulation and limit its further clinical application [11,12]. Scientists prepared polymers for improving the physical stability of polymer micelles through various cross-linking methods. Nanohydrogel is one of them. As one of the classifications of hydrogels, nanohydrogels are defined as a nano-scale polymer with 3D network nanoparticles, this network is formed by cross-linking polymer chains swelled in a good solvent [13,14]. Compared with commonly used nanocarriers (such as micelles), nanohydrogels contain a large amount of water in a swollen state, and stably load drugs, reducing the loss of drugs during blood circulation.

Generally speaking, hydrogels can be divided into two types according to cross-linking methods, one is a chemical crosslinking hydrogel [15], the other is a physical crosslinking hydrogel [16]. The chemical crosslinking hydrogel is usually crosslinked by chemical crosslinking agent and the force of cross-linking is the covalent bond generated by the functional groups between the polymer chains, so it has better mechanical strength and stability than physically cross-linked hydrogels [17,18]. However, the addition of a chemical crosslinking agent may lead to biodegradable hydrogels become a huge problem [19]. Physical cross-linking hydrogels mainly use intermolecular hydrogen bonding as the main cross-linking force, avoiding the formation of undegradable bonds, and has better biocompatibility, degradability and application prospect than chemical cross-linked hydrogels. The polylactic acid stereocomplex has many advantages. Due to its extraordinary biological, thermal, and mechanical properties, it shows great potential as a biological material. The group of Sytze J. Buwalda conducted research to prepare PEG-PLA star-shaped block copolymer hydrogels through physical crosslinking combined with photopolymerization. The gel is degraded by the hydrolysis of the ester groups in the PLA chain, resulting in the loss of physical and chemical crosslinks. This study shows that the injectable, photocrosslinked PEG-PLA star block copolymer hydrogel is a promising material for biomedical applications such as long-term controlled drug delivery [20].

The broad application prospects of stimulus responsive hydrogels have led to various single-response [21], double-response and multi-response hydrogels being studied and applied [22,23,24]. However, there are few reports on reduction/temperature dual response hydrogels. In recent years, biodegradable hydrogels with degradable bonds have become a hot topic [25,26]. Scientists have introduced degradable bonds such as hydrazine [27], enzymes [28,29], disulfide bonds [30], etc. into polymer molecules to achieve the synthesis of degradable hydrogels. Among them, disulfide bonds are favored by scientists because they can be cleaved and reformed under relatively simple and mild conditions [31,32]. In this article, we first synthesize an initiator 2-((2-Hydroxyethyl) disulfanyl) ethyl methacrylate (HSEMA) with disulfide bond through the monoesterification reaction of 2, 2′-dithiodiethanol and methacryloyl chloride, and then used it for ring-opening polymerization of poly(lactide) to form macromonomer HSEMA-PLLA and HSEMA-PDLA, the temperature-sensitive monomers 2-(2-methoxyethoxy) ethyl methacrylate (MEO_2_MA) and oligo (ethylene glycol) methacrylate (OEGMA) undergo free radical polymerization with macromonomer HSEMA-PLLA or HSEMA-PDLA under the condition of AIBN as the initiator to form temperature/reduction dual responsive micelles. Mixing equal amounts of PLA enantiomeric micellar solution in situ generates a temperature reduction dual stimulus responsive gel. The physically cross-linked gel network generated by the self-recombination of micelles and then stereo-complexation provides a safe and practical new method for the preparation of temperature/reduction dual responsive physical cross-linking gel.

## 2. Materials and Methods

### 2.1. Materials

L-lactide, D-lactide (99.0%) and methacryloyl chloride (95%, *M*_n_ = 104.53 g·mol^−1^) were purchased from Macleans (Shanghai, China). Oligo (ethylene oxide) methacrylate (OEGMA, 95%, *M*_n_ = 475 g·mol^−1^) and 2-(2-methoxy ethoxy) ethyl methacrylate (MEO_2_MA, 95%, *M*_n_ = 188.22 g·mol^−1^) were obtained from TCI (Shanghai Development Co., Ltd., Shanghai, China). 2,2′-dithiodiethanol (90%, *M*_n_ = 154.25 g·mol^−1^) was purchased from Alfa Aesar (Shanghai, China). 1,4-dithiothreitol (DTT, 99.0%, *M*_n_ = 154.25 g·mol^−1^) was purchased from Rhawn (Shanghai, China), Tin(II) 2-ethylhexanoate (99%, *M*_n_ = 405.12 g·mol^−1^)was purchased from J&k (Beijing, China), 2,2′-Azoisobutyronitrile(AIBN) ( Macleans, Shanghai, China) was purified by recrystallization from methanol. The solvents used are all dried and double distilled water is used in aqueous solutions.

### 2.2. Synthesis of 2-((2-Hydroxyethyl) Disulfanyl) Ethyl Methacrylate (HSEMA)

2-((2-Hydroxyethyl) disulfanyl) ethyl methacrylate (HSEMA) was synthesized by the monoesterification reaction of 2,2′-dithiodiethanol and methacryloylchloride [33,34]. The synthesis route was as follows: 2,2′-dithiodiethanol (4 mmol, 0.542 mL) and triethylamine (1 mL) were completely dissolved in 20 mL dried THF. Then methacryloyl chloride (4 mmol, 0.407 mL) was added drop-wise for 20 min in an ice bath at 0 °C, the reactants were stirred at room temperature for 24 h. The produced white solids, i.e., triethylamine salt, were removed using the method of filtration. Subsequently, the solvent was removed by rotary evaporation, and the product was purified by column chromatography.

### 2.3. Synthesis of Macromonomer HSEMA-PDLA

Synthesis of macromonomer HSEMA-PLLA/HSEMA-PDLA used the method of ring-opening polymerization with HSEMA as the initiator and lactide as the raw material. The synthesis route was as follows: D-lactide (6.9 mmol, 1 g), initiator HSEMA (0.53 mmol) and dry toluene (15 mL) were added to a dry and clean three-necked flask (Figure 1). After the mixture were fully dissolved, the catalyst Tin (II) 2-ethylhexanoate (Sn(Oct)_2_) (55 μL) was added. The reaction system was refluxed at 120 °C for 24 h under a nitrogen atmosphere, and then the crude products were precipitated and purified twice in cold anhydrous ether. Finally, the products were stored in a vacuum drying cabinet at room temperature. The synthesis of HSEMA-PLLA was similar to HSEMA-PDLA.

### 2.4. Synthesis of Graft Copolymer P(MEO_2_MA-co-OEGMA)-g-(HSEMA-PLLA)

The graft copolymer P (MEO_2_MA-co-OEGMA)-g-(HSEMA-PLLA) was synthesized by free radical polymerization with 2-(2-methoxy ethoxy) ethyl methacrylate (MEO_2_MA) and oligo (ethylene oxide) methacrylate (OEGMA) as temperature-sensitive monomers, HSEMA-PLLA as macromonomers and 2,2′-Azoisobutyronitrile (AIBN) as the initiator. The synthesis route was as follows: the macromonomer HSEMA-PLLA and DMF (3 mL) were added to a dry and clean Shrek tube. After HSEMA-PLLA were fully dissolved, MEO_2_MA, OEGMA and initiator AIBN were added to the mixed solution. The reaction system was reacted at 70 °C for 6 h under nitrogen atmosphere. The crude products were precipitated and purified twice in ice anhydrous ether, then dialyzed at room temperature for 3 days. Finally, the products were freeze-dried and stored in a vacuum drying cabinet at room temperature.

### 2.5. Synthesis of Temperature/Reduction Nanogels

Copolymers of equal mass were respectively dissolved in water to obtain a uniform single enantiomeric graft copolymer solution. Then the enantiomeric graft copolymer solutions were mixed in equal volumes at 37 °C to obtain a stereocomplexed gel [35,36]. The complex mechanism was shown in Figure 1.

### 2.6. Characterization of the Samples

#### 2.6.1. Nuclear Magnetic Resonance Spectroscopy (^1^H-NMR)

The structure of the compound was characterized using a Bruker-300MHz spectrometer (300 MHz Avance, Bruker Corporation, Karlsruhe, Germany) with CDCl_3_ as a solvent and tetramethylsilane (TMS) as an internal standard at room temperature.

#### 2.6.2. Fourier Transform Infrared Spectroscopy (FT-IR)

The functional group of the compound was characterized by Fourier transform infrared spectroscopy (FTIR) (Tensor 27, Bruker Corporation, Karlsruhe, Germany). Before the measurement, the sample was uniformly mixed with dry KBr, dried in a vacuum drying oven for 24 h, and finally pressed at room temperature.

#### 2.6.3. The Wide-Angle X-ray Diffraction (WAXD)

The wide-angle X-ray diffraction (WAXD) analysis was performed on Ni-filtered Cu Kα (λ = 0.154 nm) at 25 °C on the D8 Advance diffractometer from Bruker, Karlsruhe, Germany, with a 2*θ* scanning rate of 2°/min and a scanning range of 5–40°.

#### 2.6.4. Low Critical Solution Temperature (LCST)

The LCST of the gels were measured by UV-V is spectroscopy (TU-1901, Beijing Purkinje General Instrument Corporation, Beijing, China). A series of nanogel solutions (2.00 mg/mL) was arranged in a transparent sample vial. The transmittance of the nanogels was measured using the TU-1901 at a wavelength of 660 nm, photometric mode, and a temperature range of 20–50 °C.

#### 2.6.5. The Scanning Electron Microscope (SEM) Analysis

The gels samples were immersed in the distilled water at 25 °C till swell-equilibrium. The swollen gels were quickly frozen in liquid nitrogen and then freeze-dried in a freeze dryer. Microscopic morphology of the gel was performed on a desktop scanning electron microscope (SEM, TM 3030 Hitachi, Tokyo, Japan) in low vacuum mode. Before the observation, in order to improve the conductivity of the sample, it was subjected to gold spray treatment for 80 s.

#### 2.6.6. Dynamic Light Scattering (DLS) Analysis

Laser particle size meter (BI-90Plus, Brookhaven, New York, United States) was used to measure the particle size of nanogels under different conditions: 1 mg/mL of solution was prepared and filtered by 0.45 μm water filter. The particle size was measured at room temperature and variable temperature, respectively.

#### 2.6.7. Swelling Kinetics of the Gel

The preparation method of the dried gel used for the swelling performance test is: 0.5 g enantiomeric copolymer was put in a sample bottle respectively, add THF right amount to dissolve completely. In an ice bath, the THF solution of the copolymer was dripped drop by drop into a round-bottomed flask containing 5 mL of secondary water. Ultrasound at 4 °C until the solution reaches equilibrium (about 30 min), distilled THF in the solution under reduced pressure at room temperature to obtain a uniform enantiomeric graft copolymer micelle solution with a mass fraction of 10 *w*/*v*%, after mixing the same amount of enantiomeric copolymer solution and stirring for 40 min, ultrasonic for 30 min to complete the complexation, heat up in the temperature range of 25–70 °C and volatilize the solvent and obtain a completely dry PLA-based gels.

The swelling kinetics and de-swelling kinetics of the gel were explained by the weighing method. The specific method of the swelling kinetics is as follows: fully dried gels were soaked in the secondary water at 25 °C, taken out, the mass was weighed three times after a certain period of time, and the water on the surface of the gel was absorbed by filter paper. Taking the average of the data, its swelling ratio was calculated according to the following formula.
(1)$$[\mathrm{Swelling}\;\mathrm{ratio}]=\left(W_{t}-W_{d}\right)/W_{d}$$
where W_d_ is the mass of dry gel (g), W_t_ is the mass of the gel taken out at a fixed time point during the swelling process (g).

#### 2.6.8. Transmission Electron Microscopy (TEM) Analysis

The morphology of the gel was observed by a Field Emission Transmission Electron (JEOL JEM-2100, Tokey, Japan) with accelerating voltage 200 kV. The samples were prepared by placing a drop of 1.00 g·mol^−1^ copolymer solution on a carbon-coated copper grid (200 mesh) and then drying in the vacuum at 25 °C.

#### 2.6.9. Determination of Disulfide Bond Reduction of Sulfhydryl in Nanohydrogel

We use the Ellman reagent to characterize the redox responsiveness of the hydrogel. Preparation of Ellman reagent is made using a buffer solution with pH = 8 as the solvent for the Ellman reagent which was specifically prepared by weighing 1.56 g of sodium phosphate and 0.0372 g of EDTA and dissolving the volume to 100 mL, and then using sodium carbonate to adjust the pH to 8. Dissolve 4 mg of 5,5′-dithiobis (2-nitrobenzoic acid) in 1 mL pH = 8 buffer solution, store in the dark, and prepare it for immediate use; the Ellman reagent was successfully prepared. Take 4 mL of the prepared 1 mg/mL nanohydrogel solution, add the reducing agent tris(2-carboxyethyl)phosphine (TCEP) hydrochloride (1 mL, 0.02 mol/L) and Ellman’s reagent (0.2 mL), and let stand for 20 min [37]. At the same time, use the same method to treat without reducing as a blank comparison, the solution to be tested needs to be diluted 10 times to ensure that the concentration of sulfhydryl groups is within the measurement range. Use anultraviolet-visible spectrophotometer to measure within the measurement range of 200–500 nm.

## 3. Results and Discussion

The graft copolymer P (MEO2MA-co-OEGMA)-g-(HSEMA-PDLA) was synthesized by radical polymerization of 2,2-Azoisobutyronitrile (AIBN) as initiator and macromonomer HSEMA-PDLA, 2-(2-methoxy ethoxy) ethyl methacrylate (MEO2MA), oligo (ethylene oxide) methacrylate (OEGMA) (relevant data as shown in Table 1), then the same amount of enantiomeric graft copolymer micellar solution were mixed in situ to form nanogel. The amount of temperature-sensitive monomer and macromonomer used in the experiment is the same. The variable set is the degree of polymerization of the macromonomer, i.e., the amount of lactide added when synthesizing the macromonomer is different (relevant data as shown in Table 1).

### 3.1. Structural Characterization of the Initiator, Macromolecular Monomers, and Branching Copolymers

The initiator 2-((2-Hydroxyethyl) disulfanyl) ethyl methacrylate (HSEMA) was mainly synthesized by the monoesterification reaction of 2,2-dithiodiethanol and methacryloyl chloride, and Et_3_N acts as an acid binding agent to promote the reaction to move forward. The obtained crude product was purified by column chromatography (silica gel, eluent: ethyl acetate/petroleum ether = 1.5:1). Figure 1 shows the infrared spectrum of the initiator HSEMA. It can be seen from the figure that the peak at 3432 cm^−1^ is -OH stretching vibration, and 2944–2874 cm^−1^ is –C-H stretching vibration of -CH_2_- and -CH_3_. The peak at 1707 cm^−1^ is -C=O stretching vibration, at 1633 cm^−1^ is -C=C- stretching vibration, 1451–1385 cm^−1^ is -CH_3_ bending vibration, and 1155 cm^−1^ is stretching vibration of –C-O-C-. Figure 2a is the proton nuclear magnetic resonance spectrum of HSEMA, as shown in the figure, ^1^H-NMR (300 MHz, CDCl_3_) δ ppm: 6.11, 5.58 (s, 2H, CH_2_=C(CH_3_)-), 4.40 (t, 2H, -OCH_2_CH_2_-S-S-), 3.87 (t, 2H, -CH_2_OH), 2.95, 2.87 (t, 4H, -CH_2_-S-S-CH_2_-), 1.93 (s, 3H, CH_2_=C(CH_3_)-. Combining the corresponding data of the above, the synthesis of HSEMA is successful.

The synthesis of macromonomer was achieved by ring-opening polymerization of lactide with HSEMA as initiator and Tin (II) 2-ethylhexanoate (Sn(Oct)_2_) as catalyst. Figure 2b is the proton nuclear magnetic resonance spectrum of the macromonomer HSEMA-PDLA. ^1^H NMR (300 MHz, CDCl_3_) the corresponding chemical shift is δ ppm: 6.06, 5.52 (s, 2H, CH_2_=C(CH_3_)-), 5.13–5.06 (m, 1H, -(C=O)CH(CH_3_)O-), 4.30–4.35 (m, 4H,-OCH_2_CH_2_SS-), 2.91–2.82 (m, 4H, -S-S-CH_2_CH_2_O-), 1.88 (s, 3H, CH_2_=C(CH_3_)-), 1.52 (d, 3H, -(C=O)CH(CH_3_)O-). Figure 3b is the FT-IR spectrum of the macromonomer HSEMA-PDLA. As shown in the figure, the peaks at 3631 cm^−1^ and 3026–2871 cm^−1^ are the stretching vibrations of -OH and -CH_2_-CH_3_ respectively. The peaks at 1758 cm^−1^ and 1187 cm^−1^ are the stretching vibration of -C=O and –C-O-C- respectively, and the peak at 1450 cm^−1^ is the bending vibration of -CH_2_, -CH_3_. Since the disulfide bond is in the FT-IR fingerprint region. Therefore, combining the above data, the macromonomer HSEMA-PDLA was successfully synthesized, and the relevant data of HSEMA-PLLA are basically the same as HSEMA-PDLA, and no explanation is given here.

Figure 2c is the 1H-NMR spectrum of the graft copolymers (1H-NMR (300 MHz, CDCl_3_)), the specific chemical shift is: 5.12–5.06 (m, 1H, -(C=O)CH(CH_3_)O-), 4.04 (s, 2H, -O(CH_2_CH_2_O)2CH_3_), 3.5 (d, 2H, -O(CH_2_CH_2_O)_2_CH_3_), 3.33 (s, 3H, CH_2_=C(CH_3_)-),1.52–1.49 (d, 3H, -(C=O)CH(CH_3_)O-), 1.19 (s,3H,-CH_2_C(CH_3_)CH_2_-), 0.83 (s,2H, -CH_2_C(CH_3_)CH_2_-). Figure 3a is the FT-IR spectrum of the gels. It can be seen from the figure that the C=O stretching vibration peak of the gels at 1729 cm^−1^ shifts to a lower wavelength compared to the C=O (1758 cm^−1^) of the macromonomer. This is due to the complex hydrogen bonding between the PLA groups, which makes the C=O shift, it also shows that the stereocomplex gels were successfully synthesized.

### 3.2. Analysis of Complex Structure

We used a wide-angle X-ray diffraction (WAXD) to confirm the stereocomplexation of the gels in a macroscopic state. Figure 4a is the WAXD pattern of the macromonomer (red line) and gel4 (black line). As shown in the figure, the macromonomerHSEMA-PLLA_25_ has the isomorphous peaks at 2*θ* = 17°, 19° and gel4 has a relatively obvious stereocomplex peak at 2*θ* = 12°. According to the relevant literature, the synthesis of stereocomplex gel is successful [38]. Figure 4b is the WXRD pattern of the gels with different PLA polymerization degrees. It can be seen from the figure that the stereocomplexed gels with different PLA polymerization degree have different degrees of peaks at 2*θ* = 12°, 21° and 24°. where 2*θ* = 12°, 21° is more obvious and the diffraction peaks are obviously enhanced with the increase of PLA polymerization degree, this is because the increase of PLA polymerization degree increases the content of stereocomplex and the intensity of complex peak increases.

### 3.3. Micellar Properties of Hydrogels

The addition of temperature-sensitive monomer MEO_2_MA and OEGMA makes the synthesized nanogels temperature sensitive. Figure 5 shows a digital camera diagram of gels 1–4 at different temperatures. As shown in photos, gels 1–4 show different clarity at room temperature due to different hydrophobic chain lengths. and become turbid with increasing hydrophobic chain segments. When the temperature rises to 35 °C, the nanogels become further turbid. As the temperature rises further, the turbidity increases and the clarity decreases further, this is because with the increase of temperature, the hydrogen bond between the hydrophilic part of the nanogel and the water molecule weakens. The enhanced hydrophobic effect makes the solution turbid, this also means that the gel is temperature sensitive.

By adjusting the ratio between the temperature-sensitive monomers MEO_2_MA and OEGMA, the low critical solution temperature (LCST) can be adjusted to near the physiological temperature of the human body. In this study, the content of the hydrophilic part is fixed, and the variable is the degree of polymerization of the hydrophobic section of PLA. As shown in Figure 6, due to the different content of the hydrophobic part, it has different light transmittance at 25 °C, from largest to smallest, 96%, 77%, 56%, 45%, as the degree of polymerization of the hydrophobic part increases, the light transmittance gradually decreases. This is because when the content of hydrophilic chains in the polymer molecules is the same, the more the content of hydrophobic chains, the greater the degree of phase separation and the more turbid and the lower the light transmittance; Due to the same content of the hydrophilic part, the LCST values of nanohydrogels with different hydrophobic polymerization degrees are different with different hydrophobic polymerization degrees. The LCST values of gel1, gel2, gel3, and gel4 are 35 °C, 34 °C, 33 °C, 30 °C, and the LCST value decreases with the increase of the degree of polymerization of the hydrophobic part. This is because when the content of the hydrophilic part of the same, the higher the degree of polymerization of the PLA, the higher the content of the hydrophobic moiety, the lower temperature required when the hydrophobic interaction formed between the polymers is broken. In other words, the corresponding lower critical solution temperature is lower.

### 3.4. Analysis of the Particle Size and Micro Morphology of Nanogel in Solution

Nanogels are realized by self-recombination of an equal amount of enantiomeric graft copolymer solution, so nanogels still exist in the form of micelles [35]. DLS can be used to measure the particle size of nano-scale polymers in different states. Figure 7 is a graph of the particle size of nanohydrogels with different degrees of polymerization of polylactic acid. As shown in the figure, when the degree of polymerization of the hydrophobic part is different, the particle sizes of the corresponding nanohydrogels are also different. When the degree of polymerization increases, the particle size of the corresponding nanohydrogel is 150 nm, 174 nm, 255 nm, and 345 nm in sequence. As the degree of polymerization increases, the corresponding particle size increases in sequence. This is because the structure of the nanohydrogel is still a self-reorganized micellar structure of the polymer, which is composed of a hydrophobic core and a hydrophilic shell. At the same hydrophilic content, the greater the degree of polymerization of the hydrophobic part of the polymer, the larger the hydrophobic core of the corresponding micelle, so that the particle size increases with the increase of the degree of polymerization of polylactic acid.

The morphology of the nanohydrogel can be analyzed bytransmission electron microscope. Figure 8 is the transmission electron micrographs of the nano hydrogel under different conditions. To have more intuitive data on the particle size, we use dynamic light scattering (DLS) to corroborate the corresponding particle size. As shown in the figure, the particle size of gel1 at room temperature is about 150 nm, while the particle size at 37 °C is about 100 nm. This is because in the core-shell structure of the nanohydrogel, the hydrophilic shell shrinks as the temperature increases due to its temperature sensitivity, which reduces the particle size of the nanohydrogel; The particle size after reduction with 10 mM DTT is about 270 nm, which is larger than the particle size at room temperature. This is because the addition of the reducing agent causes the disulfide bond connecting the hydrophobic segment in the nanohydrogel to be reduced to a sulfhydryl group and breaks. The number of broken free hydrophobic groups increases, and aggregation is more likely to occur, forming larger aggregates to increase the particle size of the nanohydrogel. The change of particle size under different conditions also shows that the synthesized nanohydrogel has dual stimulus responsiveness of temperature and reduction.

### 3.5. Determination of Reduced Sulfhydryl Group by Ultraviolet-Visible Spectroscopy

The disulfide bond in the nanohydrogel gives it the characteristic of reduction response. When it is used as a drug carrier, it can be targeted for release under the reduced environment of tumor cells. There are many detection methods for disulfide bonds, among which Ellman reagent detection is one of the detection methods widely used in many fields such as biochemistry. The main detection mechanism is to first reduce the disulfide bond to a sulfhydryl group with a strong reducing agent, and then use the color reaction of the Ellman reagent with the free sulfhydryl group and the special peak of the sulfhydryl group in the ultraviolet spectrum to measure, therefore further confirming the reduction responsiveness of the gel. Figure 9 is the UV spectrum of the hydrogel in the initial state and the reduced state, as shown in the figure, compared with the spectrum of the initial nano hydrogel without reducing agent, the reduced nanohydrogel has a characteristic peak of free sulfhydryl at 412 nm, and the reduction conversion rate is greater than 98% calculated according to the formula of absorbance and concentration. This also shows that the disulfide bond in the nanohydrogel is successfully reduced to a sulfhydryl group, i.e., the nanohydrogel has reduction responsiveness.

### 3.6. Analysis of Swelling Kinetics of Hydrogel

Swelling is one of the methods to measure the performance of the gel. To illustrate the water absorption performance of the hydrogel, a swelling kinetics experiment was carried out using a fully dried hydrogel. Figure 10 shows the change of the swelling ratio of the gel with time at 25 °C. As shown in the figure, the gels of different polymerization degrees have different swelling ratios, and the swelling equilibrium is reached at about 210 min. Among them, the swelling ratio of gel1 is the largest, about 2.45, and the swelling ratio of gel4 is the smallest, about 0.95. This is because the hydrophilic segment P (MEO_2_MA-*co*-OEGMA) in the gel can combine with water molecules to form hydrogen bonds, gel1 has the lowest degree of polymerization of PLA, the physical crosslinking point was the least and the three-dimensional complexation was the weakest, thus forming a relatively loose three-dimensional network of gels with a high swelling ratio. In gel4, PLA has the highest degree of polymerization and the strongest three-dimensional complexation, so the gel structure formed is more compact and the swelling ratio is low.

Figure 11 is the physical image and scanning electron microscope image of gel before and after swelling. The size of each square in the background is 1 cm. It can be seen from the figure that the volume of gel increases significantly after swelling with water. Combined with the scanning electron microscope image, it can be seen that gel has an obvious three-dimensional network structure.

### 3.7. Rheological Analysis

The rheological behavior is an important method to study gels. We conduct the gelation process and mechanics of complex gels through rheological experiments on equal amounts of nanohydrogels at the same concentration (10 *w*/*v*%). As we all know, *G’* and *G”* represent the storage modulus and loss modulus of the material respectively. When *G**’* > *G**”*, the elastic deformation is dominant and the solution is in a gel state. When *G′* < *G”*, the viscosity deformation is dominant, and the solution is in a sol state. When *G′* = *G”*, the corresponding independent variable is the sol-gel transition point of the micellar solution. Among them, the storage modulus *G′* related to the mechanical strength of the gel. The larger the storage modulus is, the greater the mechanical strength of the corresponding gel is [39]. Figure 12 shows the analysis of the gelation process and the oscillation stress scanning diagram of the gel at 37 °C. As shown in Figure 12a, gelation time of gel2 and gel3 is different due to their different PLA block lengths, the gelation time is 101 s and 58 s respectively. This is because the greater the degree of PLA polymerization, the more PLLA or PDLA groups on the grafted chain, which increases the physical cross-linking points, reduces the gelation time, and makes the three-dimensional complexation more obvious. To further verify the impact of the degree of polymerization on the mechanical properties of the gel, we performed a stress scan on the resulting gels. Figure 12b shows the oscillation stress scan curve of the gel. As shown in the figure, different polymerization degrees gels have different yield values (the corresponding stress values when *G′* = *G″* in the figure). The corresponding yield values of gel2 and gel3 are 64 Pa and 70 Pa, respectively, and the storage modulus of gel2 is always higher than gel3 before the yield value. This is because the increase in the degree of PLA polymerization increases the physical crosslinking points of the gel, and the resulting gel. The tighter the structure, the better the mechanical properties of the gel.

## 4. Conclusions

A novel type physically crosslinked nanogel was synthesized by self-recombination of enantiomeric micellar solution under mild conditions and in situ stereocomplexation. Thermosensitive monomers MEO_2_MA and OEGMA make it thermally responsive. The introduction of the disulfide bond (-S-S-) makes it reduction responsive and can be degraded within two weeks. This was proven in DLS experiments. The existence of stereocomplexation in the hydrogel was confirmed by WAXD. A rheometer was used to study the gelation process and mechanical strength, and it was confirmed that it was related to the degree of PLA polymerization. In the case of the same hydrophilic part content, the influence of the degree of polymerization of the hydrophobic segment (iepolylactic acid) on the micelle properties, swelling properties and rheological behavior of the nanohydrogel was studied. Experimental results show that the performance of nanohydrogels can be adjusted by changing the degree of polymerization of polylactic acid; at the same time, ultraviolet spectroscopy and dynamic light scattering are used to verify the reduction response and temperature-sensitive response of nanohydrogels. The preparation of physical crosslinking temperature reduction dual responsive hydrogel provides a new method.

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
