# Peer review of "Temperature/Reduction Dual Response Nanogel Is Formed by In Situ Stereocomplexation of Poly (Lactic Acid)"

_polymers, 2021, doi:10.3390/polym13203492_

Round 1

Reviewer 1 Report

The manuscript (polymers-1379776) entitled "In-Situ Formation of Temperature/ Reduction Dual Responsive Nanogels by Stereocomplexation of Poly(Lactic Acid)" provides sound discussion with a reasonable set of experiments. Although the manuscript needs to be revised considering the following suggestions for further improvement in its quality.

  1. The title needs to be a little modified for more clear understanding.
  2. The abstract should be revised by incorporating specific results and key findings and novelty. The present form of the abstract is more generalized.
  3. Add a similar line of research carried by other investigators in the introduction section and highlight their key findings. 
  4.  Section 2.6. need to divide into sub-section according to different characterization techniques utilized in characterization. 
  5. Section 3.1 should be divided into two separate sections, one as synthesis section and another as characterization section.
  6. It is suggested to include the Standard deviation (SD) in particle size results and mention polydispersity index (PDI) results as well.
  7. Correct the caption of figure 12. some detail missing. 
  8. The formatting error in the conclusion section needs to be rectified. 

Author Response

请参阅附件。

Reviewer 2 Report

Stimulus responsive polymers and hydrogels are gaining more popularity for biomedical use. The authors conduct a detailed, well-planned study to evaluate the effects of macromonomer hydrophobicity on nanogel behavior and swelling. However this work needs further characterization to evaluate how the gels perform when loaded with a drug. Ring opening polymerization should not be abbreviated on page 2, line 95. Why was the sample mixed into the KBr disc rather than sandwiched between two KBr discs? Were the NMR spectra integrated to determine % yield of each of the synthesized chemicals? If so, what was the % yield for the initatior, the macromonomer, and the graft co-polymer? Were there any drug-loading studies conducted on the different gel formulations to evaluate the loading concentration and release profile?

Author Response

请参阅附件。

Round 2

Reviewer 2 Report

The appropriate changes have been made to the manuscript. Please address the question below, however the manuscript is acceptable for publication. 

1. How was yield determined for each step of the synthesis? What is the overall yield of the nanogels?
